# Mangiferin and Hesperidin Transdermal Distribution and Permeability through the Skin from Solutions and Honeybush Extracts (*Cyclopia* sp.)—A Comparison Ex Vivo Study

**DOI:** 10.3390/molecules26216547

**Published:** 2021-10-29

**Authors:** Anna Hering, Jadwiga Renata Ochocka, Helena Baranska, Krzysztof Cal, Justyna Stefanowicz-Hajduk

**Affiliations:** 1Department of Biology and Pharmaceutical Botany, Medical University of Gdansk, 80-416 Gdansk, Poland; 2Department of Pharmaceutical Technology, Medical University of Gdansk, 80-416 Gdansk, Poland; helena.baranska@gmail.com (H.B.); krzysztof.cal@gumed.edu.pl (K.C.)

**Keywords:** skin penetration, skin permeation, HPLC, Fabaceae, honeybush, fluorescent microscopy

## Abstract

Polyphenolic compounds—mangiferin and hesperidin—are, among others, the most important secondary metabolites of African shrub *Cyclopia* sp. (honeybush). The aim of this study was to compare the percutaneous absorption of mangiferin and hesperidin from solutions (water, ethanol 50%, (*v/v*)) and extracts obtained from green and fermented honeybush (water, ethanol 50%, (*v/v*)). Research was performed with the Bronaugh cells, on human dorsal skin. The mangiferin and hesperidin distributions in skin layers (stratum corneum, epidermis, and dermis) and in acceptor fluid (in every 2, 4, 6, and 24 h) were evaluated by HPLC–Photodiode Array Coulometric and Coulometric Electrochemical Array Detection. The transdermal distribution of hesperidin was also demonstrated by fluorescence microscopy. Results indicated that mangiferin and hesperidin were able to cross the stratum corneum and penetrate into the epidermis and dermis. An advantage of hesperidin penetration into the skin from the water over ethanol solution was observed (451.02 ± 14.50 vs. 357.39 ± 4.51 ng/cm^2^), as well as in the mangiferin study (127.56 ± 9.49 vs. 97.23 ± 2.92 ng/cm^2^). Furthermore, mangiferin penetration was more evident from nonfermented honeybush ethanol extract (189.85 ± 4.11 ng/cm^2^) than from solutions. The permeation of mangiferin and hesperidin through the skin to the acceptor fluid was observed regardless of whether the solution or the honeybush extract was applied. The highest ability to permeate the skin was demonstrated for the water solution of hesperidin (250.92 ± 16.01 ng/cm^2^), while the hesperidin occurring in the extracts permeated in a very low capacity. Mangiferin from nonfermented honeybush ethanol extract had the highest ability to permeate to the acceptor fluid within 24 h (152.36 ± 8.57 ng/cm^2^).

## 1. Introduction

Natural products used in the form of cosmetics or nutraceuticals nowadays are gaining importance. They are usually well known in ethnomedicine as a relatively cheap and safe source of active compounds [1]. There are many synthetic products on the market that are dermatologically effective, though they cause unwanted side-effects. The most commonly observed are redness, irritation, rashes, and itching. In addition, such products cannot be often used in the area of injured skin, on mucosa, or in high amounts, due to the toxic systemic effects [2]. Exposure to biotic and abiotic stress generates oxidative stress among skin macromolecules and, in consequence, their dysfunction or degradation [3,4,5]. Due to the fact that skin is considered the most external and first barrier of the human body, its arrangement is essential to maintain internal homeostasis. The first layer—the stratum corneum, is composed of dead, keratin-filled cells. However, they contain substances with hygroscopic properties, responsible for binding water, and enzymes participating in metabolic processes, the activity of which is responsible for the acidic pH of the skin surface. The specific arrangement of the stratum corneum and the hydrophobic properties create this impenetrable barrier layer to pathogens and foreign substances [3,6,7]. Penetration through the stratum corneum barrier is the basic parameter in studies of the transdermal distribution of xenobiotics [8,9]. With the growing interest in the percutaneous route of administration of medicinal substances as an alternative to oral intake, intensive research has been started on increasing the permeability of the stratum corneum layer [7]. The main problems of those studies are: determining whether the introduced substance should target the dermis or reach the bloodstream; physical and biological properties of a xenobiotic; the drug partition between cumulation in the skin and permeation through the skin; individual differences and skin diseases [9,10,11,12]. Considering polyphenols as almost perfect antioxidants, which can be used to protect the skin and its macromolecules against oxidative stress [13,14], their ability to penetrate into and permeate through skin layers should be under consideration. The permeability depends, among others, on the structure of the molecules and their chemical properties; however, the transdermal distribution of these compounds could be modified by the composition of the delivery system [15,16].

Polyphenols are a group of pharmacologically active, secondary metabolites, widely spread in nature. It is confirmed that polyphenols have ability to protect skin macromolecules from internal and external degradation. Natural sources of specific polyphenolic compositions combined with high quantitative content are in demand [1].

*Cyclopia* spp. (Fabaceae) is an endemic shrub growing wild and cultivated on plantations only within Cape Province in the Republic of South Africa. Leaves and branches from honeybush are collected and dried (green honeybush) or fermented and dried (fermented honeybush), to prepare the tea beverage with many health properties [17]. The richness in polyphenolic compounds, low tannins, and lack of caffeine have made this sweet-tasting tea beverage a highly attractive alternative for *Camellia sinensis* tea [18,19]. Additionally, extracts from honeybush are traditionally used by the local people in different skin diseases [17]. The tisane is becoming increasingly more popular in commercial use among consumers, and researchers.

The pharmacological activity of *Cyclopia* spp. is caused by the synergic effect of polyphenolic fraction—xanthones, flavanones, flavones, isoflavones, coumestans, phenolic acids, and others [20]. Therefore, extracts from honeybush have strong antioxidant properties with a large amount of mangiferin and hesperidin, compounds well known as oxidative stress fighters [17,20,21,22,23,24,25,26]. These compounds, among others, are also responsible for the skin UV protection [20,27] and inhibition of wrinkle formation [28]. In addition, Bartoszewski et al. indicated that mangiferin and hesperidin are safe for keratinocytes [29].

Mangiferin, a C-glucosyl xanthone, is widely distributed in nature. It is presented abundantly in the families Anacardiaceae [30], Liliaceae [31], or Fabaceae [18]. The many important healing pharmacological activities of mangiferin include analgesic, immunomodulatory, antibacterial, and antidiabetic [32,33,34]. One of the major properties of mangiferin is the lowering level of oxidative stress, which promotes in the long-term many degradation diseases [35,36,37]. Despite numerous in vitro analyses confirming mangiferin activity, its bioavailability is limited by both low solubility and absorption from the intestine [38,39]. Among the new routes of improving mangiferin bioavailability is topical application. It was confirmed that mangiferin has the ability to pass through the stratum corneum, penetrate the epidermis and dermis, and inhibit the degradation of collagen and elastin, as well as wrinkle formation [40,41]. Additional results have confirmed mangiferin’s ability to interact with macromolecules of the skin and increase the regenerative capacity of wounded skin [42]. Mangiferin has a high potential in dermatology; however, many aspects of the compound action in the skin have not yet been investigated.

Hesperidin, a flavanone (hesperetin-7-*O*-rutinoside), is abundantly presented in citrus fruits [43,44]. Mostly known from cardiovascular protection and anti-inflammatory functions, hesperidin has proved significant systemic effects, which are limited by the low water solubility and low absorption from the intestine [45,46,47,48,49,50,51]. In recent years, dermatological studies have focused on hesperidin’s ability to improve the skin functions and condition [52]. The transdermal activity of hesperidin is mainly related to its antioxidant properties. It has been confirmed that hesperidin catalyses the scarring process through the reduction of oxidative stress around the wounded area [52,53], and enhances the ability to protect keratinocytes against UV-induced damage [54,55,56]. It has also been proven that hesperidin inhibits the inflammatory reaction generated by keratinocytes when exposed to oxidative stress [57]. In addition, hesperidin, by reducing free radicals generated in the course of the melanogenesis process, inhibits melanogenesis and melanosome transport, resulting in skin lightening [58,59,60]. The topical application of hesperidin improved skin barrier permeability [61]. What is important is that the compound has no toxic effects after neither oral nor topical application, whereas many ingredients present in cosmetic products may cause allergic reactions and be responsible for adverse systemic effects [52,62,63,64,65]. Until now, animal models or tissue cultures have been used in studies of hesperidin, while the transdermal distribution of this polyphenol after application on human skin has never been analysed.

In this study, we estimated the permeation and distribution of mangiferin and hesperidin among skin layers ex vivo after the application of two types of solutions (water and ethanol) and *Cyclopia* sp. extracts (water and ethanol). The results of *Cyclopia* sp. extracts, as well as hesperidin permeation and transdermal distribution, were shown in this work for the first time.

## 2. Results

Extracts A, B, C, and D (Table 1), as well as water or ethanol solutions of hesperidin or mangiferin, were applied on the skin surface. After 2, 4, 6, and 24 h, samples of acceptor fluid were taken to quantitatively analyse both compounds. The estimation of hesperidin and mangiferin distribution among skin layers was determined after 24 h of experiment, followed by skin layers’ separations and extractions. For the quantitative determination of mangiferin and hesperidin in water and ethanol extracts applied on the skin (summarised in Table 1), and in solutions extracted from individual layers of the skin, as well as in acceptor fluids, HPLC analysis with electrochemical detection was utilised.

The ex vivo experiments revealed that mangiferin [40] and hesperidin from solutions (water or ethanol) or extracts (water or ethanol) are capable of passing through the stratum corneum and penetrating into the deeper layers of the skin, as well as permeating through the skin.

### 2.1. Hesperidin Permeation into the Skin from Solutions and Honeybush Extracts

The ability of hesperidin to cross the skin barrier—the stratum corneum—has been estimated and demonstrated in Figure 1 and Table 2.

The quantitative analysis of hesperidin distributed among skin layers after its application from solutions indicated the advantage of hesperidin permeation both from water and ethanol solutions. The highest amount of hesperidin was detected in the dermis in the case of both solutions, and it was almost four times higher than in the stratum corneum. The results were 225.38 ± 29.72 and 185.59 ± 0.45 ng/cm^2^ for the water and ethanol solutions, respectively. The HPLC analysis also showed that the hesperidin concentration was almost two times higher in the epidermis than in the stratum corneum for both the solutions (Table 2).

The advantage of hesperidin cumulation among skin layers after the application of solutions compared to honeybush extracts (A, B, C, and D) was evidenced. The amount of hesperidin in the stratum corneum, epidermis, and dermis was higher for water than for ethanol solution.

In the experiments with the extracts, the lowest amount of hesperidin was for ethanol extract B in the stratum corneum. The concentration of hesperidin in the epidermis after honeybush extracts were applied on the skin surface was comparable with a slight predominance of hesperidin cumulation from extract D. Among the dermis, the differences in hesperidin cumulation were more visible: the lowest hesperidin concentration was detected for extracts C and D, whereas the hesperidin cumulation after the application of extracts A and B exceeded 100 ng/cm^2^. Both for the solutions and extracts, the highest concentration of hesperidin was detected in the dermis (Figure 1 and Table 2).

### 2.2. Hesperidin Permeability through the Skin from Solutions and Honeybush Extracts

HPLC analysis revealed hesperidin’s ability to permeate through the skin to the acceptor fluid (Figure 2).

The permeation of hesperidin after application on the skin surface of hesperidin solutions—(HeH_2_O, HeEtOH)—was similar and within the limits of the statistical error in the first 2 h. The quantified analysis showed a high amount of hesperidin presented in acceptor fluid after 2 h from application, which indicated the relatively quick penetration of the compound through the skin. A further increase in the amount of hesperidin was observed after 4 and 6 h. After 24 h, the HPLC analysis showed an amount of hesperidin in the fluid at least twice higher compared to the amount determined after 2 h (Table 2).

Hesperidin permeation through the skin, after the application of honeybush extracts on the skin surface, was also observed, and it increased during the time of incubation. After 24 h of hesperidin, the amount in the acceptor fluid (calculated on cm^2^ of the skin) was highest for extract D (22.41 ± 1.56 ng/cm^2^). For the remaining extracts, the obtained amounts of the compound in the fluid decreased as follows: C > A > B.

### 2.3. Transdermal Distribution of Hesperidin with the Use of Fluorescence Microscopy

Fluorescence microscopy was utilised for confirmation of the transdermal distribution of hesperidin after 24 h of incubation of the ethanol solution on the human skin surface ex vivo. Followed by completion of the permeation experiment, transverse skin cuts were made. For microscopic analysis, only the pieces of skin with all visible layers of the skin were used. Freshly taken cross-cuttings of skin were analysed with a fluorescence microscope (Nikon Eclipse 50, filter UV 2A, Ex 330–380 nm). The results in the form of photographs are presented in Figure 3. Comparative observation of the skin with pure ethanol (control, Figure 3A) and ethanolic hesperidin solution applied on the skin showed a strong, bright yellow-green fluorescence in the analysed sample (Figure 3B). According to the literature data, the observed fluorescence colour is characteristic for hesperidin [66]. The intensity of fluorescence was low in the area of the stratum corneum and epidermis, while the dermis layer showed clearly visible yellow-green fluorescence. In this image (Figure 3B), the strong fluorescence presented on the external surface of the skin indicated that some part of hesperidin did not penetrate into the stratum corneum, and it was not completely washed away after the end of the experiment. This hesperidin was probably deposited on the surface of the skin.

The complex structure of the skin and its macromolecules determine its auto-luminescence, which is visible in the left image (Figure 3A, skin sample on which only ethanol was applied, control sample) [67]. The obtained pictures indicate that hesperidin enhances skin fluorescence.

The fluorescence microscopy results confirm the quantitative analysis of the transdermal distribution of hesperidin after its application and incubation on the skin surface (Table 2). The fluorescence was noticeable especially in the area of the dermis where, according to the results presented in Figure 1, hesperidin underwent the highest absorption. However, according to Xie et al., hesperetin, the aglycon of hesperidin, interacts with proteins, affecting with slight fluorescence the quenching of macromolecules [68]. An interaction could be caused by both hydrophobic and electrostatic bonds between proteins and hesperetin. It was indicated that hydroxyl groups are the most important in the proton transversion [69]. Hesperetin, which interacts with proteins, can be stored and floated with them [68]. Our comparison data of the control and intense yellow-green fluorescence presented in the right picture (Figure 3B) suggest that nonbonded hesperidin was responsible for the bright fluorescence. A probably unknown amount of hesperidin interacted with macromolecules of the epidermis or dermis, resulting in quenching. However, the amount of noninteracted hesperidin was still enough to give visible fluorescence in the dermis. The free hesperidin presented in the dermis layer is needed to protect ECM macromolecules from internal and external oxidative stress [14].

### 2.4. Comparison of Mangiferin Permeation into the Skin and Permeability through the Skin from Solutions and Honeybush Extracts

The ability of mangiferin to cross the skin stratum corneum barrier was estimated and demonstrated in our previous paper [40] where we analysed both the mangiferin distribution among skin layers and the permeation through the skin to the acceptor fluid, after application of mangiferin solutions (MH_2_OH, MEtOH) on the skin surface. In this study, we additionally estimated the distribution and permeation of the honeybush extracts (A, B, C, D) through the skin.

HPLC analysis with electrochemical detection indicated that mangiferin is capable of entering into and crossing through the stratum corneum, after the application of both mangiferin solutions and honeybush extracts (Table 3 and Figure 4).

The highest amount of mangiferin that entered into and permeated through the stratum corneum is presented for extract C, which is the most promising source of mangiferin for skin permeation. The compound accumulation was highest in the epidermis, while, in the stratum corneum and dermis, the amount of determined mangiferin was comparable (Table 3). The second significant amount of mangiferin in transdermal distribution was observed after the application of MH_2_O (Table 3 and Figure 4). In this case, the mangiferin cumulation was highest in the dermis.

The concentration of mangiferin detected in the stratum corneum was similar to each other with only slight differences: MH_2_O > MEtOH > B > D > A (Table 3). However, the highest amount was detected for extract C. In the epidermis, the lowest concentration of mangiferin was estimated for extracts A and D and ethanol solution MEtOH, while mangiferin from extract B and MH_2_O indicated a more significant accumulation of the compound. In the dermis, differences in mangiferin concentration were as follows: C > MH_2_O > MEtOH > B > D = A. The lowest accumulation of mangiferin was detected among skin layers after the application of extracts A and D.

The mangiferin permeation through the skin from all analysed solutions and honeybush extracts was constant and occurred relatively quickly (Figure 5). The most effective permeation of mangiferin into the acceptor fluid, during the 24 h of analysis, was observed for extract C, while the least was for MH_2_O, extracts A and B. The mangiferin permeation process for extracts A and B was similar, and a slight increase was observed for the MEtOH solution and extract D.

## 3. Discussion

The first stage in the process of xenobiotic penetration into the skin and permeation through the skin is overcoming the stratum corneum barrier, which is treated as a semi-permeable membrane. Both analysed compounds in our study—mangiferin and hesperidin—have logP between 1 and 3 (hesperidin: 1.78, mangiferin: 2.73); the mangiferin molar mass does not exceed 500 Da (422.34 g/mol), though the hesperidin molar mas is higher with the value of 610.19 g/mol. This suggests that mangiferin and hesperidin should be able to pass the stratum corneum [12,70,71]. The limiting factor for both polyphenols could be their low water solubility and a branched structure [72,73]. Our analysis revealed the ability of mangiferin and hesperidin to permeate the stratum corneum layer from both honeybush extracts and solutions, with a predominance of hesperidin from the water solution and mangiferin from the ethanol extract of green honeybush (extract C). Both compounds presented a relatively fast ability to permeate through the skin, especially from applied solutions for hesperidin and extract C for mangiferin. These solutions and the extract applied on the skin exhibited the highest cumulation of analysed compounds among skin layers, which make them the most promising sources of mangiferin and hesperidin release into the skin and through the skin.

Surprisingly, the lowest hesperidin amount from both solutions was accumulated in the stratum corneum, and the highest amount was detected in the dermis. Our analysis showed that hesperidin sourced from the water solution penetrated into the skin and permeated through the skin with an advantage over hesperidin from the ethanol solution. These data could indicate that for hesperidin, ethanol is not a sorption promoter [74]. Despite the lower hesperidin solubility in water [72], a higher amount of this polyphenol can permeate to the dermis after the application of a water solution on the skin. The dermis, as the largest layer of the skin, is composed of sensitive to oxidative stress proteins, and the presence of hesperidin, a compound with strong antioxidant properties, can work as effective protection against macromolecules [52].

The confirmation of hesperidin cumulation among skin layers and the interaction of hesperidin molecules was observed with fluorescence microscopy. The experiment confirmed that a lower yellow-green fluorescence assigned to hesperidin could be seen in the stratum corneum and epidermis, while the distribution of the xenobiotic in the dermis was brighter. The intensity of fluorescence in the epidermis and dermis after hesperidin application proved the presence of free hesperidin, which did not interact with skin macromolecules [68]. The permanent hesperidin interaction with molecules of the dermis extracellular matrix (ECM) was generally slight, though the mechanism of the protection can differ from the conventional antioxidant trail [75,76,77]. Nevertheless, the experiment was carried out ex vivo, and it should be emphasised that in a living organism, the partition of free hesperidin and that associated with macromolecules could be different. The ECM is the largest element of healthy skin and is responsible for homeostasis and the preservation of the mechanical and biological properties of the skin. It is highly complex system, the protection of which by antioxidants is thus essential [21,68].

It should be emphasised that fluorescence techniques are used to determine transdermal drug uptake, though they could be utilised only in the case of molecules with fluorescence properties that have different wavelengths than skin macromolecules [40,67,78,79]. As it was indicated in our research, hesperidin with its fluorescence properties can be successfully analysed with these techniques [66]. However, the visualisation of the transdermal distribution of hesperidin from honeybush extracts was impossible due to a large number of different polyphenols with fluorescence abilities presented in the extracts [17].

The present study also included the analysis of hesperidin permeation from the skin after the application of hesperidin solutions during the 24 h. The research indicated that hesperidin occurred in the acceptor fluid relatively quickly, especially after the first two hours. During the following hours, hesperidin permeation was constantly occurring despite the solution used, with the advantage of water solution. The differences between the permeability of hesperidin derived from water and ethanol solution were seen especially after 24 h of the analysis. The experimental data obtained in this part confirmed that hesperidin from water solution has a better ability to penetrate into the skin and permeate through the skin than hesperidin can from ethanol solution.

We also analysed hesperidin permeation to the skin and through the skin after application of the honeybush extracts on the skin surface. The experiments showed the best accumulation of hesperidin in the dermis, also from the extracts obtained with water. These results indicate that, regardless of the solution or extracts applied on the skin, hesperidin permeates in a higher amount to the dermis from the water carrier. There is also a slight advantage in the penetration into the skin of hesperidin from water-unfermented extracts compared to fermented honeybush (A > B).

A major disadvantage of using plant material is the high variability in chemical composition and antioxidant properties, which depends not only on the changing environmental conditions, but also on the time of harvest, as well as storage and transport conditions [20,80,81,82]. If we consider polyphenolic fractions as a source of antioxidants in the skin layers, it is important to estimate “competition” and “enhancer” properties of chemicals [15,16]. Honeybush extracts contain many biologically active molecules, which could potentially improve or inhibit the penetration of the most quantitative compounds into the skin: mangiferin and hesperidin [83]. In the case of both compounds, no comparison analysis of the cumulation into the skin or permeation through the skin from solutions and plant extracts has been performed so far.

Despite the applied solutions or honeybush extracts on the skin surface, mangiferin presented a lower ability than hesperidin to penetrate into the skin. Ethanol extract from the green honeybush (extract C) applied on the human skin surface showed the highest mangiferin penetration into the skin and the fastest ability to permeate to the acceptor fluid during the 24 h of conducted analysis. There was also no evidence of an advantage in accumulation and permeation studies of mangiferin from solutions than mangiferin from honeybush extracts. The water solution of mangiferin, similar to hesperidin solution, indicated a higher ability for transdermal distribution than ethanol solution. Our study showed that mangiferin has a limited ability to pass the stratum corneum barrier and to obtain higher concentrations among skin layers.

However, the above results of mangiferin permeation from honeybush extracts (especially from nonfermented ethanol—C) are generally promising, especially considering the fact that more mangiferin was absorbed from the ethanol extract than from solutions with higher concentrations of the compound. The obtained data of mangiferin permeation do not give an unambiguous answer that ethanol could be a sorption promoter for mangiferin delivered into the skin from honeybush extracts. The synergism of biological activity of polyphenolic compounds in this plant species may be responsible for this effect [84,85]. In addition, the strengthening action of hesperidin on mangiferin has been previously observed [29,83].

In conclusion, mangiferin and hesperidin were detected in all skin layers and in the acceptor fluid in the form of both solutions and extracts applied on the human skin surface. The analysis revealed the predominance of the transdermal cumulation of hesperidin over mangiferin. On the other hand, permeation through the skin was better for mangiferin than for hesperidin. Our ex vivo study on mangiferin showed that its permeation into and through the skin is significantly higher for the ethanol-nonfermented extract than for solutions. The hesperidin transdermal distribution and permeation study on *Cyclopia* sp. extracts are described in this work for the first time. The results clearly indicated that hesperidin solutions and mangiferin, originating from the ethanol extract of “green” honeybush, can be a source of transdermal release of the analysed molecules, which have strong protective and regenerative properties in the skin. The compounds may be used as antioxidants in dermatology and in topical ways of drug administration, especially due to the limited oral absorption of both polyphenols.

## 4. Materials and Methods

### 4.1. Materials

Mangiferin, hesperidin, and sodium azide were purchased from Sigma Chemical Co. (St. Louis, MO, USA). HPLC-grade methanol and ethanol were purchased from P.O.Ch. (Gliwice, Poland).

### 4.2. Plant Material and Extract Preparation

Plant material was composed of dried leaves and branches declared as the mixtures of different honeybush species (*Cyclopia* sp.). Nonfermented (“green”) *Cyclopia* sp. material was purchased from Ukajali Marcin Majka Co., (Krakow, Poland) for preparation of extracts A and C. Fermented *Cyclopia* sp. plant material was purchased from KAWON-HURT Co., (Gostyn, Poland) for preparation of extracts B and D. Plant material and used solvents are summarised in Table 1. Representative samples of each plant material were deposited at the Department of Biology and Pharmaceutical Botany, Medical University of Gdansk, Poland. Extract preparation: 15 g of each plant material was mechanically homogenised to obtain a particle size < 2.0 mm. Extraction with 100 mL of water or ethanol (50%, *v/v*) was performed three times with the use of an ultrasonic water bath (50 Hz) for 30 min at 90 °C. After filtration through Whattman filter paper (390 µm pore size), obtained extracts were evaporated under vacuum at 60 °C. The dry residue was redissolved in either water or 50% (*v/v*) ethanol to the final concentration of 30 mg/mL. For the quantitative analysis of mangiferin and hesperidin in the resulting extracts, HPLC with Photodiode Array Coulometric and Coulometric Electrochemical Array Detection was used.

### 4.3. Penetration and Permeation Studies

The studies were performed with the use of human cadaver full skin. Before the experiment, the skin was collected from the region of the thorax of six 40–60-years-old donors, and stored frozen at −20 °C. The solutions and extracts were applied as an infinite dose: mangiferin and hesperidin, and ethanol (50%, *v/v*) and water solutions at concentrations of 25 μg/mL and 4 μg/mL, respectively; and extracts A–D at a concentration of 30 mg/mL. The ethanol at a concentration of 50% (*v/v*) was applied on the skin as a blank control. The skin (thickness—0.75 mm; diffusion area—0.65 cm^2^) was placed in the flow-through Bronaugh diffusion cell apparatus (Figure 6). The static diffusion cell is composed of two chambers: donor and acceptor. Between these chambers, a part of skin is placed. The analysed sample is applied on the upper skin surface and the dermis is in constant contact with homogenised and circulated acceptor fluid. The permeant from the skin can be taken to the analysis at a specific time period.

Saline solution, 20 mL (with 0.005% sodium azide), was recirculated beneath the skin with a constant rate of 10 mL/h. Acceptor fluid ensured the sink condition. Solutions or extracts were applied on the surface of the skin and left for 24 h. The chamber system was incubated at the temperature of 37 °C ± 0.5 °C. After 2, 4, 6, and 24 h from the application of penetrants, samples of acceptor fluid were collected. Before skin layers’ disconnection, the penetrants were removed from the skin. The stratum corneum was separated by the tape-stripping method, using 30 fragments of a 3 m adhesive tape with the following parameters: pressure ~1 kg/cm^2^ (applied by stamp), 2 s duration of pressure, and a rapid removal rate at an angle of 45°. The epidermis and dermis were isolated by the heat separation technique [86]. The whole skin was immersed in water at 60 °C for 45 s; afterward, the skin layers were divided by tweezers. All skin layers were extracted with methanol and the obtained solutions were analysed by HPLC.

The analysis was under the revision and approval of the Independent Bioethics Commission for Research of the Medical University of Gdansk (number NKBBN/120-41/2014).

The permeability and penetration data were expressed as mean values ± standard deviation (SD). Statistical comparisons of the results were analysed with a two-way ANOVA with post hoc Tukey test (*p* < 0.05).

### 4.4. Chromatography

The HPLC system and analysis condition were described previously [29]. Briefly, the HPLC system was equipped with a spectrophotometric diode array 340S detector pump P 580, column thermostat STH 585, and automated sample injector ASI-100 (Dionex Corporation, Sunnyvale, CA, USA). In addition, for the results confirmation, an additional detector was utilised: Coulochem II electrochemical detector with a 5020 model guard and a 5010 model analytical cell (ESA, Chelmsford, MA, USA) operated by the Chromeleon chromatography-management system v. 6.8 (Dionex). The separation of mangiferin and hesperidin was performed on a Hypersil Gold C_18_ column (150 mm × 4.6 mm I.D., 5 μm particle) equipped with a Hypersil Gold guard column (10 mm × 4.6 mm I.D., 5 μm particle) (Thermo Electron Corporation, Dreieich, Germany).

Before use, the isocratic mobile phase (15 mM of sodium phosphate, pH 4.0 with 85% orthophosphoric acid and acetonitrile (65/35 (*v/v*)) was filtered (0.22 μm membrane filter) and degassed by vacuum. The flow rate was maintained at 1.0 mL/min and the injection volume was 20 μL. The temperature in the column was 20 °C and that in the automated sample injector was set at 8 °C. To establish the electrochemical behaviour of mangiferin and hesperidin, repeated injections of working standard solutions (0.1 mg/mL) and detection at potentials from −0.5 to +1.2 V were studied [87]. Both mangiferin and hesperidin exhibited good responses of hydrodynamic voltammograms in the ranges from +0.35 to +0.95 V. The applied potentials on the guard cell, electrode E1, and electrode E2 were +1.1 V, +0.35 V, and +0.95 V, respectively. The photodiode array detector was set at 225, 254, 280, and 360 nm wavelengths, respectively [83].

Mangiferin and hesperidin exhibited a good linearity of calibration curves in the investigated ranges (10.0–100.0 ng, R^2^ > 0.9995). Limits of detection (LOD) and quantification (LOQ) for mangiferin and hesperidin were 0.35 and 9.7, and 0.24 and 8.6 ng/mL, respectively. Precision and reproducibility were estimated from six consecutive replicates, and the R.S.D. values for mangiferin and hesperidin were lower than 0.9% and 3.7%, respectively.

### 4.5. Fluorescence Microscopy

The utilisation of fluorescence microscopy to confirm the distribution of mangiferin in the different skin layers was described by Ochocka et al., 2017 [40]. Both mangiferin and hesperidin are substances having fused heterocyclic rings in their structure, which determine their ability to fluoresce [88,89]. The excitation and emission maxima of hesperidin are 350 nm and 420 nm, respectively [66]. Biomolecules such as NADPH, elastin, and collagen also exhibit fluorescence [67], which makes observation much more difficult because interactions between chemical compounds having chromophores can enhance or quench fluorescence [90].

To confirm the transdermal distribution of hesperidin with the use of fluorescence microscopy, an additional experiment was performed. The hesperidin solution applied on the skin was 10 µg/mL (96% EtOH (*v/v*)); in the control study, only ethanol was used (96% EtOH (*v/v*)). The course of the research was analogous to that in the permeation study, but only without collecting samples from the outlet and separation of the skin layers after the experiment. Skin removed from the Bronaught cells was frozen at −60 °C in a Cryotome E (Thermo Electron Corporation, Rugby, UK). Just before fluorescence microscopy, several cross-cuttings of the frozen skin were performed in order to obtain the transverse profile of the skin. Sections with all visible layers of the skin profile were placed on a laboratory glass slide. Freshly thawed skin cuttings were analysed with a fluorescence microscope (Nikon Eclipse 50 with filter UV 2A: 330 nm excitation, 380 nm emission, equipped with the lamp, Nikon super-high-pressure mercury, and the camera, Nikon 05–5MC). The analysis of the obtained images (magnification 100 times) was performed in the program NIS Elements AR3.2.

## Figures and Tables

**Figure 1 molecules-26-06547-f001:**
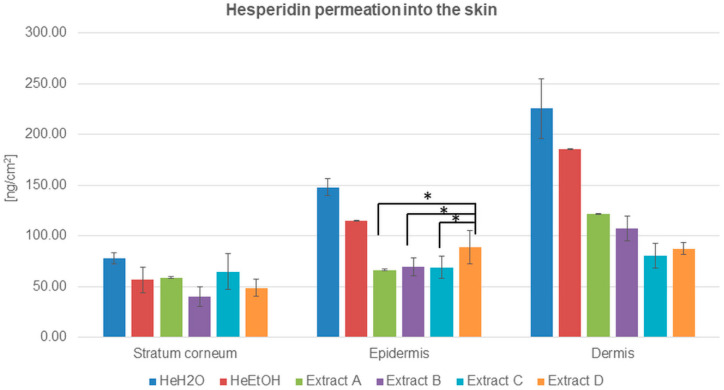
Hesperidin distribution among skin layers: stratum corneum, epidermis, and dermis (ng/cm^2^) after application and 24 h of incubation with hesperidin solutions: water (HeH_2_O, 4 μg/mL), ethanol (HeEtOH, 4 μg/mL, 50% (*v/v*)), or honeybush extracts: A, B, C, D. Each average value was obtained from three independent repetitions. Error bars represent standard deviations. Significant differences among samples are marked with an asterisk (*p* < 0.05).

**Figure 2 molecules-26-06547-f002:**
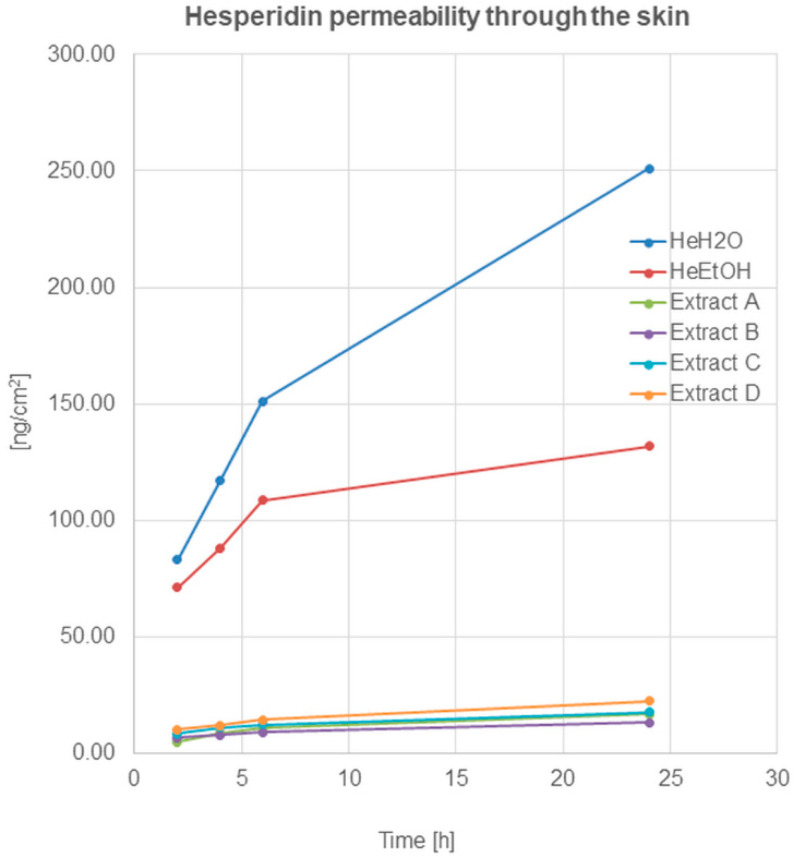
Hesperidin permeation through the skin after 2, 4, 6, and 24 h (ng/cm^2^) from the application of hesperidin solutions: water (HeH_2_O, 4 μg/mL), ethanol (HeEtOH, 4 μg/mL, 50% (*v/v*)), or honeybush extracts: A, B, C, D.

**Figure 3 molecules-26-06547-f003:**
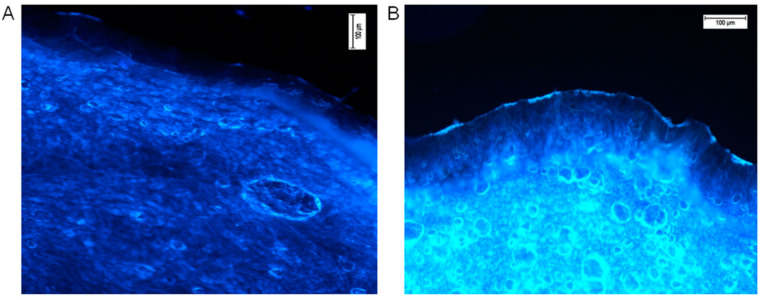
Hesperidin distribution in the human skin (**A**—the control, ethanol 96% alone, **B**—ethanol hesperidin solution 96%, 0.01 mg/mL). Ethanol and hesperidin solutions were applied on the human skin ex vivo. After 24 h of incubation, skin was washed in water and analysed under a fluorescence microscope, Nikon Eclipse 50, filter UV 2A, Ex 330–380 nm.

**Figure 4 molecules-26-06547-f004:**
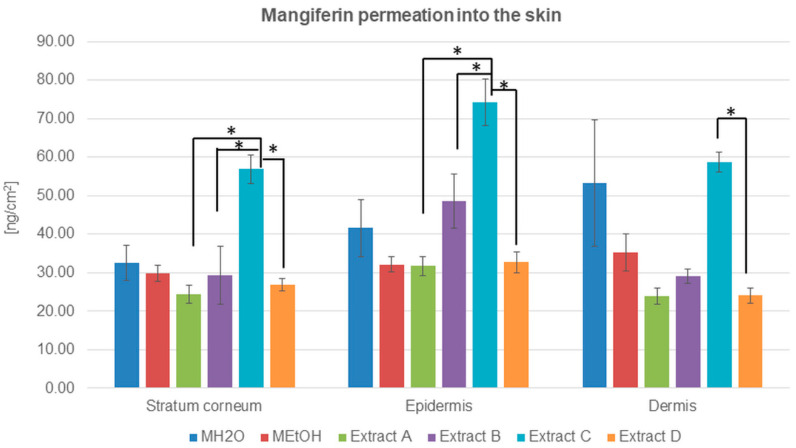
Mangiferin distribution among skin layers: stratum corneum, epidermis, and dermis (ng/cm^2^) after application and 24 h of incubation of mangiferin solutions: water (MH_2_O, 25 μg/mL), ethanol (25 μg/mL, MEtOH, 50% (*v/v*)), or honeybush extracts: A, B, C, D. Each average value was obtained from three independent repetitions. Error bars represent standard deviations. Significant differences among samples are marked with an asterisk (*p* < 0.05).

**Figure 5 molecules-26-06547-f005:**
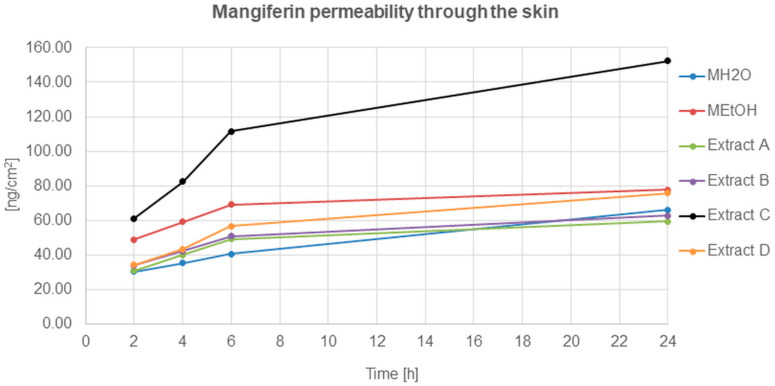
Mangiferin permeation through the skin after 2, 4, 6, and 24 h (ng/cm^2^) from the application of mangiferin solutions: water (MH_2_O, 25 μg/mL), ethanol (MEtOH, 25 μg/mL, 50% (*v/v*)), or honeybush extracts: A, B, C, D.

**Figure 6 molecules-26-06547-f006:**
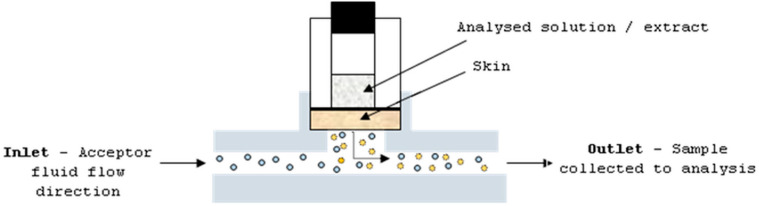
Diffusion cell apparatus.

**Table 1 molecules-26-06547-t001:** Summary of content of mangiferin and hesperidin [ng/mL] in *Cyclopia* sp. extracts analysed by HPLC-UV.

*Cyclopia* sp. Extract	Type of Plant Material	Solvent for the Extracts Preparation	Mangiferin	Hesperidin
**A**	nonfermented (“green”)	H_2_O	1593.1 ± 31.08	483.6 ± 16.74
**B**	fermented	H_2_O	967.2 ± 22.17	376.13 ± 7.07
**C**	nonfermented (“green”)	50% EtOH (*v/v*)	4695.68 ± 65.33	423.98 ± 13.01
**D**	fermented	50% EtOH (*v/v*)	2962.05 ± 62.14	304.5 ± 9

**Table 2 molecules-26-06547-t002:** Accumulation of hesperidin analysed by HPLC-UV and HPLC-EC among skin layers: stratum corneum, epidermis, and dermis, and hesperidin permeation through the skin after 2, 4, 6, and 24 h (ng/cm^2^) from the application of hesperidin solutions (HeH_2_O, HeEtOH) or honeybush extracts (A, B, C, D).

	HeH_2_O	HeEtOH	Extract A	Extract B	Extract C	Extract D
**S.c.**	77.74 ± 5.44	56.77 ± 12.59	59.03 ± 0.84	40.21 ± 9.9	64.82 ± 17.39	48.87 ± 8.68
**Epidermis**	147.9 ± 8.35	115.03± 0.5	66.56 ± 0.9	69.54 ± 8.51	68.92 ± 11	88.85 ± 16.68
**Dermis**	225.38 ± 29.72	185.59 ± 0.45	121.85 ± 0.56	107.33 ± 12.56	80.21 ± 12.05	87.44 ± 5.78
**2 h**	83.18 ± 12.79	71.33 ± 12.38	5.08 ± 1.22	6.72 ± 1.84	8.43 ± 0.83	10.31 ± 3.06
**4 h**	117.18 ± 12.79	88.13 ± 20.04	8.87 ± 2.35	8 ± 1.35	11.18 ± 0.7	12 ± 2.91
**6 h**	151.08 ± 22.35	108.77 ± 14.38	10.97 ± 0.38	9.13 ± 0.38	12.51 ± 3.91	14.31 ± 2.91
**24 h**	250.92 ± 16.01	132 ± 14.93	16.72 ± 4.22	13.23 ± 4.24	17.79 ± 3.84	22.41 ± 1.56

S.c. stratum corneum; each average value was obtained from three independent repetitions.

**Table 3 molecules-26-06547-t003:** Accumulation of mangiferin analysed by HPLC-UV among skin layers: stratum corneum, epidermis, and dermis, and mangiferin permeation through the skin after 2, 4, 6, and 24 h (ng/cm^2^) from the application of mangiferin solutions (MH_2_O, MEtOH) or honeybush extracts (A, B, C, D).

	MH_2_O	MEtOH	Extract A	Extract B	Extract C	Extract D
**S.c.**	32.62 ± 4.61	29.92 ± 2.07	24.46 ± 2.36	29.44 ± 7.51	56.92 ± 3.72	26.92 ± 1.53
**Epidermis**	41.62 ± 7.45	32.08 ± 1.97	32.74 ± 2.45	48.51 ± 7.08	74.26 ± 5.95	32.72 ± 2.74
**Dermis**	53.32 ± 16.41	35.23 ± 4.73	23.95 ± 2.15	29.13 ± 1,84	58.67 ± 2.67	24.1 ± 1.97
**2 h**	30.23 ± 5.22	48.85 ± 7.37	30.97 ± 1.54	33.85 ± 5.31	61.03 ± 5.05	34.1 ± 7.44
**4 h**	35.38 ± 5.9	59.08 ± 6.76	39.95 ± 1.72	42.31 ± 8.18	82.36 ± 7.27	43.28 ± 6.74
**6 h**	40.54 ± 7.77	69.15 ± 7	49.23 ± 2.66	50.92 ± 7.52	111.54 ±9.39	56.56 ± 4.66
**24 h**	66.15 ± 11.92	77.78 ± 8.38	59.33 ± 3.76	62.82 ± 9.46	152.36 ± 8.57	75.69 ± 2.14

S.c. stratum corneum; each average value was obtained from three independent repetitions.

## Data Availability

All data are included in the manuscript.

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
