# Peer review of "Mangiferin and Hesperidin Transdermal Distribution and Permeability through the Skin from Solutions and Honeybush Extracts (Cyclopia sp.)—A Comparison Ex Vivo Study"

_molecules, 2021, doi:10.3390/molecules26216547_

Round 1

Reviewer 1 Report

Cyclopia spp. (Fabaceae family). what is the rational behind using this plant? is the plant widespread distributed or mainly present in south Africa?

Why did the authors perform Etoh 50% extraction? several extraction methods and solvent are available and use in the literature (acetone, alcool based at different percentages)

why did the authors used region of the thorax for ex vivo trials?

Did the authors test 50%Etoh toxicity on the cells?

page 10, 4.3 : what do the authors mean for infinite dose?

page 11: I expect a better description of tissue cell apparatus

Author Response

Dear Editor,

We would like to thank for critical reading this manuscript and valuable suggestions. We have carefully considered all of the suggestions and made the appropriate corrections.

Comment 1: The Cyclopia spp. (Fabaceae family). what is the rational behind using this plant? is the plant widespread distributed or mainly present in south Africa?

Response: Thank you for this comment. Cyclopia spp. grows only in South Africa, though it is very popular in many countries around the world and used as pro-health tisane. This information we have added in the text: lines 62-70.

Comment 2: Why did the authors perform EtOH 50% extraction? several extraction methods and solvent are available and use in the literature (acetone, alcohol based at different percentages)

Response: EtOH as a solvent for plants extraction is more safe when applied on the skin than other popular solvents. Also solubility of mangiferin and hesperidin is far weaker in lower percentage of EtOH than 50%. Ethanol in the percentage of 50% (v/v) is commonly used to prepare herbal extracts for the dermal use due to the fact that the higher percentage could cause dryness of the skin.

Comment 3: Why did the authors used region of the thorax for ex vivo trials?

Response: During the time of our experiment only thorax parts of the skin were available from prosectorium. Furthermore, this part of human skin is usually less damage than other parts of body skin.

Comment 4: Did the authors test 50% EtOH toxicity on the cells?

Response: During the experiments: penetration to the skin and permeation from the skin EtOH 50% alone was used as a blank probe (please see the added information in lines: 398-399), giving no changes on the used human skin. EtOH 50% is commonly used to prepare herbal extracts for the dermal use.   

Comment 5: Page 10, 4.3 : what do the authors mean for infinite dose?

Response: Infinite dose, when the diffusion cells are used, corresponds to the thermodynamic activity of the permeant (analysed compound). Even if the dose of this compound is large, the diffusion of the compound is constant.

Comment 6: I expect a better description of tissue cell apparatus

Response: The apparatus is used to research in vitro and ex vivo skin penetration. It is composed of donor and acceptor chambers. Between those chambers a part of a skin is placed, with the dermis in contact with acceptor fluid, that is in a constant circulation. The analysed sample is applicated on the skin surface and the permeant from the skin could be taken to the analysis at a specific time period. The apparatus is settled in 37oC to imitate in vivo conditions. This information we have added in the text: lines: 401-405.

We thank the Reviewer for all suggestions and hope that the revised manuscript is now appropriate for publication in Molecules.

Sincerely,

Anna Hering, Ph.D.

Department of Biology and Pharmaceutical Botany,

Medical University of Gdansk, Gdansk, Poland

Reviewer 2 Report

The authors studied mangiferin and hesperidin transdermal distribution and permeability through the skin. It is an excellent study and I recommend to publish this work in Molecules.

Small text corrections:

In page 2 please change “The main problems of those study….” to “The main problems of those studies….”

In page 8 line 3, might be typo mistake please change “slight increase we observed” to “slight increase was observed”  

Author Response

Dear Editor,

We would like to thank for critical reading this manuscript and valuable suggestions. We have carefully considered all of the suggestions and made the appropriate corrections.

Comment 1: In page 2 please change “The main problems of those study….” to “The main problems of those studies….”

Response: This correction we have added in the text: line 49.

Comment 2: In page 8 line 3, might be typo mistake please change “slight increase we observed” to “slight increase was observed”

Response: This correction we have added in the text: line 261.

We thank the Reviewer for all suggestions and hope that the revised manuscript is now appropriate for publication in Molecules.

Sincerely,

Anna Hering, Ph.D.

Department of Biology and Pharmaceutical Botany,

Medical University of Gdansk, Gdansk, Poland

Reviewer 3 Report

In this paper, the author investigated the transdermal distribution and permeability of mangiferin and hesperidin through skin from solution and extracts of honeybush. Mangiferin and hesperidin are two most important metabolites from honeybush and claimed to be beneficial to reduce oxidative stress, but low absorption from intestine. Results indicated that both metabolites were able to penetrate into epidermis and dermis. This study was the first investigation of the permeation and transdermal distribution of these two polyphenols after applied on human skin.  

Questions:

  • In the title of Table 1, please adjust all the words in the font size
  • It would be helpful to mention in the title of Table 2 that the data in the Table 2 are generated by HPLC analysis. 
  • It would be much helpful if the author could clearly mention that which set of data/table/figure was generated by using either HPLC-UV or HPLC-EC methods for both hesperidin and mangiferin quantitation. 
  • In the Figure 4, the mangiferin from extract C showed highest in all stratum corneum, epidermis, and dermis. And in the Figure 6 and the paragraph beneath it, it described the experiment design and procedures. If assumed same amount of analyte applied on all the experiments, does that mean the analyte from extract C flow out the least amount from outlet compared to others? Also, it mentioned that skins were heated up in waters and before divided into different layers, I'm wondering whether analyte would lose during this process.
  • In the end of this manuscript, please remember to fill in all the  information needed from the authors.

Author Response

Dear Editor,

We would like to thank for critical reading this manuscript and valuable suggestions. We have carefully considered all of the suggestions and made the appropriate corrections.

Comment 1: In the title of Table 1, please adjust all the words in the font size.

Response: The title of Table 1 is now in one font size and the Table content is in font 9, as in the case of Table 2 and Table 3.

Comment 2 and 3: It would be helpful to mention in the title of Table 2 that the data in the Table 2 are generated by HPLC analysis. It would be much helpful if the author could clearly mention that which set of data/table/figure was generated by using either HPLC-UV or HPLC-EC methods for both hesperidin and mangiferin quantitation. 

Response: This information has been added to the titles of the Tables. Table 1, line 125; Table 2, line: 133, and Table 3, line: 233.

Comment 4: In the Figure 4, the mangiferin from extract C showed highest in all stratum corneum, epidermis, and dermis. And in the Figure 6 and the paragraph beneath it, it described the experiment design and procedures. If assumed same amount of analyte applied on all the experiments, does that mean the analyte from extract C flow out the least amount from outlet compared to others? Also, it mentioned that skins were heated up in waters and before divided into different layers, I'm wondering whether analyte would lose during this process?

Response: In the case of extract applicated on the skin surface, the penetration of analyte into the skin and permeation through the skin is dependent on the presence of remaining chemicals in the applied extract. If mangiferin from extract C cumulates among skin layers in the highest amount from all the analysed extracts, it does not mean that also permeates in the least amount (please see the permeation results after 24 h in Table 3). The remaining compounds in the extract C and their synergism probably caused the higher possibility for mangiferin diffusion through stratum corneum.

In the case of heating – according to the available literature data separation process is very fast and cause the destruction of a thin connective tissue between skin layers, resulting in negligible lost in analytes cumulated among epidermis and dermis.   

Comment 5: In the end of this manuscript, please remember to fill in all the  information needed from the authors.

Response: This information has been added in the end of the manuscript, lines: 477-479.

We thank the Reviewer for all suggestions and hope that the revised manuscript is now appropriate for publication in Molecules.

Sincerely,

Anna Hering, Ph.D.

Department of Biology and Pharmaceutical Botany,

Medical University of Gdansk, Gdansk, Poland